**Subject Category:**
Biology (whole organism)

ecology/evolution

thermal adaptation, climate change, copepod, sex-specific response, developmental plasticity, *Acartia tonsa*

**Author for correspondence:**
Matthew Sasaki
e-mail: matthew.sasaki@uconn.edu

# Complex interactions between local adaptation, phenotypic plasticity and sex affect vulnerability to warming in a widespread marine copepod

Matthew Sasaki[1], Sydney Hedberg[2], Kailin Richardson[3] and Hans G. Dam[1]

[1]Department of Marine Sciences, University of Connecticut, Groton, CT, USA
[2]Gustavus Adolphus College, St Peter, MN 56082, USA
[3]Savannah State University, Savannah, GA 31404, USA

  MS, 0000-0001-5560-5363; HGD, 0000-0001-6121-5038

Predicting the response of populations to climate change requires an understanding of how various factors affect thermal performance. Genetic differentiation is well known to affect thermal performance, but the effects of sex and developmental phenotypic plasticity often go uncharacterized. We used common garden experiments to test for effects of local adaptation, developmental phenotypic plasticity and individual sex on thermal performance of the ubiquitous copepod, *Acartia tonsa* (Calanoida, Crustacea) from two populations strongly differing in thermal regimes (Florida and Connecticut, USA). Females had higher thermal tolerance than males in both populations, while the Florida population had higher thermal tolerance compared with the Connecticut population. An effect of developmental phenotypic plasticity on thermal tolerance was observed only in the Connecticut population. Our results show clearly that thermal performance is affected by complex interactions of the three tested variables. Ignoring sex-specific differences in thermal performance may result in a severe underestimation of population-level impacts of warming because of population decline due to sperm limitation. Furthermore, despite having a higher thermal tolerance, low-latitude populations may be more vulnerable to warming as they lack the ability to respond to increases in temperature through phenotypic plasticity.

# 1. Introduction

Temperature has a profound effect on organismal performance [1,2]. Rapid climate warming represents a significant challenge for organisms, increasing average environmental temperatures [3] and the frequency of extreme climatic events such as heat waves [4]. Predicting organismal responses to these changes depends on our understanding of the factors affecting thermal tolerance. Acute thermal tolerance is known to be affected by phenotypic plasticity [5] and genetic differentiation [6], as well as diet, behaviour and individual sex [7–9]. Spatial variation in the thermal environment should generate adaptive differences in thermal performance between populations from different environments [2].

Copepods are arguably the most abundant metazoan on the planet [10]. Thus, they are intimately linked to all commercial fisheries and to global biogeochemical cycles [11]. Copepods occupy diverse ecological niches and habitat types, adopting a wide range of lifestyles. Because of their ecological importance, short-generation time and ability to being cultured in the laboratory, copepods are ideal candidates for studying adaptation to aquatic environments. Many copepod taxa have large geographical ranges, encompassing a large degree of variation in the thermal environment; thus, predicting their response to warming will be population-dependent and strongly influenced by specialist-generalist trade-offs in performance [12].

The climate variability hypothesis (CVH) [13,14] states that increased thermal tolerance should correspond with increased mean environmental temperature, while plasticity should evolve in response to variability in the thermal environment. These predictions are broadly supported in terrestrial and freshwater aquatic systems [15,16], but support in marine systems is limited by a paucity of studies [17–19], particularly in widely dispersed pelagic organisms. Moreover, these studies have generally not explicitly addressed the role of developmental phenotypic plasticity or sex-specific thermal performance. Both of these have been documented for a variety of taxa [8,17,20,21] including copepods [9,19,22,23]. Developmental phenotypic plasticity is likely an important mechanism organisms use to cope with variation in environmental conditions [24]. Sex-specific differences in organismal performance are also likely to be important in determining the outcomes of climate change. There is emerging evidence, for example, that sex-specific performance may play a large role in determining organismal responses to ocean acidification [25]. Combined with possible sperm limitation in copepod populations [26,27], this suggests that sex-specific differences in thermal adaptation are an important factor to consider in the determination of copepod vulnerability to warming, as well as for predictions based on the CVH.

*Acartia tonsa* is a widely distributed calanoid copepod, which dominates coastal and estuarine systems around the globe [28,29]. This species is characterized by relatively short generation times of the order of weeks [30–33]. There is distinct sexual dimorphism, with females being considerably larger than males. Body size is generally temperature-dependent in copepods [34,35]. In *A. tonsa*, mature females are typically less than 1 mm in length, with males averaging around 0.7 mm [36]. Unlike larger calanoids, *A. tonsa* does not maintain large lipid energy reserves [37]. With a geographical range covering a large latitudinal thermal gradient in the North Atlantic, this is a good model system to explore the contributions of various adaptive mechanisms to thermal adaptation [19]. Here, we examine the effects of genetic differentiation, developmental phenotypic plasticity and individual sex on thermal tolerance and body size in the copepod *A. tonsa*. Our results show that complex interactions between these variables strongly affect our ability to predict organismal responses to climate change.

# 2. Methods

Plankton samples were collected with surface tows at field sites in Groton, Connecticut, and Punta Gorda, Florida (table 1), during July and August 2017 using a 250 µm mesh plankton net and non-filtering cod end. Sea surface temperature data for both sites (table 1) were obtained from the AQUA-MODIS satellite database [38]. Both sampling locations were in shallow water (less than 2 m); thus, surface temperature data are likely a good representation of temperature throughout the water column. Connecticut represents a cool, more variable thermal environment compared with Florida, which is characterized by warm and stable temperatures. Daily temperature variation at each site is minor compared with the inter-site differences [39]. Initial laboratory populations of more than 1500 mature adults were established from collected animals. Cultures were maintained in 0.6 µm filtered

**Table 1.** Site name, geographical coordinates, mean annual temperature, mean annual maximum and mean annual temperature range for all collection locations.

| population | coordinates (latitude, longitude) | mean annual temperature (°C) | mean annual maximum temperature (°C) | mean annual temperature range (°C) |
| --- | --- | --- | --- | --- |
| Connecticut (CT) | 41.320591 N, −72.001564 W | 13.3 | 22.7 | 22.5 |
| Florida (FL) | 26.940398 N, −82.051036 W | 24.9 | 31.4 | 15.3 |

seawater under common garden conditions (salinity: 30 practical salinity units, 12 h : 12 h light : dark, 18°C) for several generations. During this time, copepods were fed ad libitum a diet of the microalgae *Tetraselmis* sp., *Rhodomonas* sp. and *Thalassiosira weissflogii*, which were semi-continuously cultured in F/2 medium (F/2 – silicate for *Tetraselmis* sp. and *Rhodomonas* sp.) under the same conditions. Cultures were maintained under these conditions for several generations before the experiments, thus minimizing the effects of previous environmental acclimation (i.e. differences in food abundance/ quality and temperature) in the field.

Body size measurements were taken for individuals from the laboratory cultures ($n = 30$ males and 30 females for both sexes from both populations). Individuals were isolated in a drop of filtered seawater and photographed using a camera attached to an inverted microscope after the water had been removed. Body lengths were measured as the length of the prosome using Image-J (https://imagej.nih.gov/ij/).

To test for the effect of developmental temperature, a fraction of the eggs from the 18°C culture were moved to 22°C to develop. All other variables were held constant. Once mature, individuals from both developmental conditions (18 and 22°C) were exposed to a 24 h acute heat stress. Individuals were carefully transferred to a microcentrifuge tube filled with 1.5 ml of filtered seawater, then transferred to heat blocks set to a constant temperature (18–38°C at 1°C intervals). Each individual experienced a single temperature. Individual survivorship was recorded after 24 h as binary data (1, survival; 0, mortality). Survivorship was determined during examination under a dissection microscope by response to stimuli or visible gut-passage movement. A total of 1717 individuals were used throughout the experiments (727 CT individuals and 990 FL individuals). Initial heat stresses were performed across the entire range of temperatures (18–38°C) in order to determine where additional heat stresses were needed for each of the populations. Therefore, different numbers of individuals were used for the two populations as the two temperature ranges differed between the populations.

All analyses were performed using the software package R v. 3.5.1 [40]. Body size measurements were analysed using a three-way ANOVA (body size ∼ population * developmental temperature * sex). A Levene's test was used to test the assumption of homogeneity of variance. A Tukey post hoc test was then used to examine pairwise differences between the various groups. To analyse the survivorship data, an initial ANOVA was run for all data (survivorship ∼ stress temperature + sex + developmental temperature + population, and all two-way interactions). Three-way and four-way interactions were excluded. ANOVAs were also run for each population separately (survivorship ∼ stress temperature * sex * developmental temperature). Thermal performance curves were estimated using logistic regressions on the data from both developmental temperatures from both populations. Because of the common garden design, differences in the performance curves between developmental conditions within a population can be attributed to developmental phenotypic plasticity, whereas differences between populations should reflect the effects of genetic differentiation. $LD_{50}$ (the temperature with 50% mortality) was calculated for each performance curve. The change in $LD_{50}$ between the two developmental conditions ($\Delta LD_{50}$) was used as a measure of the magnitude of the plastic response.

## 3. Results

### 3.1. Body size

Female copepods were always significantly larger than males, regardless of population or developmental temperature (figure 1, $p < 2.2 \times 10^{-16}$). Both male and female copepods from the CT population were significantly larger than copepods from the FL population in the 18°C developmental treatment

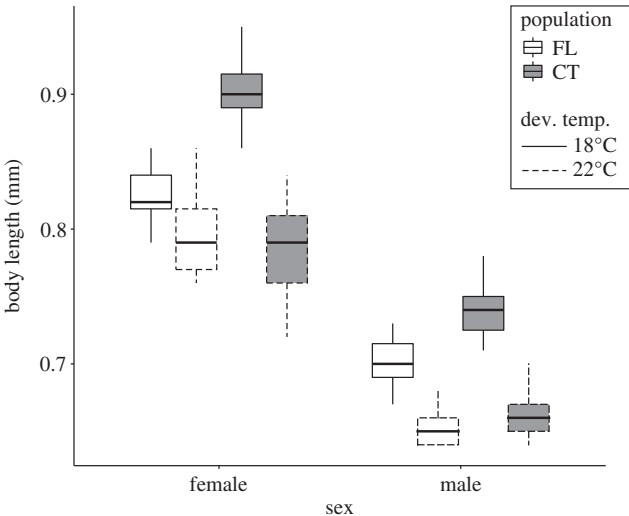

**Figure 1.** A box-and-whisker plot showing body size data for the various groups. Measurements are grouped by sex. The Florida (FL) population is shown in white, and the Connecticut (CT) population in grey. The two developmental temperature groups (18 and 22°C) are indicated with solid and dashed lines, respectively.

($p < 2.874 \times 10^{-10}$). However, there were no significant differences between the respective sexes from the CT and FL populations in the 22°C developmental treatment. Both populations saw a significant reduction in body length with an increase in developmental temperature ($p < 2.2 \times 10^{-16}$).

## 3.2. Genetic differentiation

We observed clear differences between performance curves for the two populations, consistent with a significant population effect in the full ANOVA (table 2). The Florida (FL) population performance curve was shifted towards warmer temperatures compared with the Connecticut (CT) population (figure 2). This is also reflected in the reaction norms (figure 3); FL individuals had a higher thermal tolerance than individuals of the same sex from the CT population.

## 3.3. Developmental phenotypic plasticity

The observed significant developmental temperature × population interaction term in the full ANOVA suggests that the effect of developmental temperature differed between the two populations (table 2). The ANOVA results for the CT population showed a significant effect of developmental temperature which was not observed for the FL population. Thermal performance curves for the 18 and 22°C developmental temperature treatments differed in CT but not FL individuals (figure 2, dashed versus solid lines). The slope of the reaction norms, which represents the magnitude of developmental phenotypic plasticity (figure 3), was not significantly different from zero for FL individuals, regardless of sex. By contrast, slopes of the reaction norms for both sexes in the CT population were significantly greater than zero ($p = 0.001202$).

## 3.4. Sex-dependent thermal performance traits

Males showed significantly lower survival than females in both populations (figure 2, red versus blue lines). LD$_{50}$ reaction norms also showed clear sex-dependent differences in thermal tolerance (figure 3), with females being always more tolerant than males. However, there were no sex-dependent differences in the plastic response between males and females (no Sex * Dev. Temp. interaction term in the full ANOVA), regardless of population.

# 4. Discussion

The two populations of *A. tonsa* used in this study were collected from strongly differing thermal environments—Connecticut, a cool, variable environment, and Florida, a warmer, more stable

**Table 2.** ANOVA results for the logistic regression relating survivorship to stress temperature, population, developmental temperature and individual sex. Significant terms ($p < 0.05$) are shown in italics.

| | d.f. | deviance | resid. d.f. | resid. dev | Pr(>Chi) |
|---|---|---|---|---|---|
| **all data** | | | | | |
| *stress temperature* | 1 | 928.89 | 1715 | 1409.7 | 0.00 |
| *individual sex* | 1 | 54.7 | 1714 | 1355 | $1.40 \times 10^{-13}$ |
| *developmental temperature* | 1 | 7.74 | 1713 | 1347.2 | 0.005388 |
| *population* | 1 | 17.8 | 1712 | 1329.4 | $2.45 \times 10^{-05}$ |
| *dev. temp. × population* | 1 | 5.46 | 1711 | 1324 | 0.019462 |
| sex × dev. temp | 1 | 0.27 | 1710 | 1323.7 | 0.60482 |
| sex × population | 1 | 3.82 | 1709 | 1319.9 | 0.050578 |
| *stress temp. × population* | 1 | 8.38 | 1708 | 1311.5 | 0.003801 |
| stress temp. × dev. temp. | 1 | 0.06 | 1707 | 1311.5 | 0.809781 |
| stress temp. × sex | 1 | 2.29 | 1706 | 1309.2 | 0.130522 |
| **CT model** | | | | | |
| *stress temperature* | 1 | 318.6 | 725 | 666.22 | 0.00 |
| *individual sex* | 1 | 29.42 | 724 | 636.8 | $5.82 \times 10^{-8}$ |
| *developmental temperature* | 1 | 10.49 | 723 | 626.31 | 0.001202 |
| stress temp. × sex | 1 | 0 | 722 | 626.31 | 0.960815 |
| stress temp. × dev. temp. | 1 | 0.14 | 721 | 626.17 | 0.71148 |
| sex × dev. temp. | 1 | 0.23 | 720 | 625.94 | 0.631601 |
| stress temp. × sex × dev. temp. | 1 | 1.75 | 719 | 624.2 | 0.186143 |
| **FL model** | | | | | |
| *stress temperature* | 1 | 646.63 | 988 | 706.5 | 0.00 |
| *individual sex* | 1 | 20.51 | 987 | 685.99 | $5.94 \times 10^{-6}$ |
| developmental temperature | 1 | 0.63 | 986 | 685.36 | 0.42597 |
| *stress temp. × sex* | 1 | 4.86 | 985 | 680.5 | 0.02744 |
| stress temp. × dev. temp. | 1 | 2.97 | 984 | 677.52 | 0.08465 |
| sex × dev. temp. | 1 | 0.22 | 983 | 677.31 | 0.64267 |
| stress temp. × sex × dev. temp. | 1 | 0.23 | 982 | 677.08 | 0.63395 |

environment. We observed lower thermal tolerance, but stronger plasticity in the CT population relative to the FL population, consistent with expectations of the CVH [13,14]. While the results for both male and female performance are consistent with the CVH, we find that individual sex had the largest effect on thermal tolerance. It is important to emphasize that our study is not a strong test of the CVH, as only two populations were used. However, the clear evidence that genetic differentiation, phenotypic plasticity and individual sex interact to determine thermal tolerance within the framework of the CVH is critical for our understanding of organismal responses to warming. The demographic implications of these results are crucial to consider in predictions of future population dynamics.

Large variation was also observed in the body size data. Females were always larger than males, as is commonly observed in copepods [41,42]. Females and males from the northern (CT) population were larger than their counterparts from the southern (FL) population in the 18°C developmental temperature treatment. Bergmann's rule posits that populations from higher latitudes should have larger body sizes than populations from lower latitudes [43]. Other studies of copepods have also observed body size clines in agreement with this rule [44]. Because of the common garden experimental design, the differences in body size we observed here are likely genetically determined. However, these differences are not observed in the 22°C developmental temperature group, suggesting a more complex interaction between the developmental temperature and genotype.

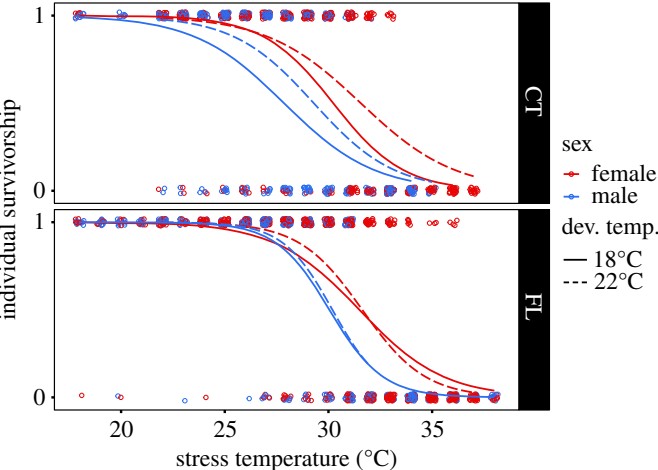

**Figure 2.** Survivorship data for adult *A. tonsa* individuals are indicated by the points (1, survival; 0, mortality). Thermal performance curves are estimated by logistic regression. Colour and line type indicate individual sex and developmental temperature, respectively.

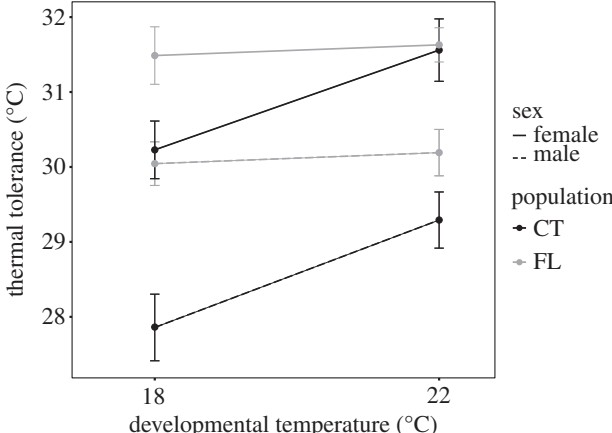

**Figure 3.** Reaction norms for adult *A. tonsa* showing thermal tolerance ($LD_{50}$) as a function of developmental temperature for both sexes (line type) from the two populations (colour). Points are thermal tolerance $\pm$ s.e. from the logistic regression models. Reaction norm slope is the magnitude of plasticity.

In both populations, females always had a higher thermal tolerance than males. Sex-specific differences in thermal tolerance are observed across diverse systems [8,9,20,21,45]. Within copepods, the few studies that have examined sex-specific thermal tolerance have also found females to be more thermally tolerant than males [9,22,23,46], but ours is the first to examine these differences in more than one adaptive mechanism (thermal tolerance and phenotypic plasticity), and in multiple populations. Interestingly, female copepods have also been found to be more tolerant to toxic dinoflagellates [47] and to starvation [48,49]. The observations of higher tolerance to diverse stressors in females may be underlain in part by a 'live fast, die young' strategy in mate-searching males [50]. Thermal tolerance is also often observed to correlate with smaller body size [43,51–53], although this appears to be strongly species-specific [54]. In our study, smaller body size is observed to correlate with increased thermal tolerance between populations, but not between the sexes; FL copepods are smaller than CT individuals and have a higher thermal tolerance while males are smaller than females but have a lower thermal tolerance.

While there are strong differences in male and female thermal tolerance in this study, neither population exhibits significant differences between male and female plastic capacity ($\Delta LD_{50}$). No previous studies have examined differences in male and female developmental phenotypic plasticity, but higher acclimatory capacity was observed in females of a different copepod species [9]. This difference in sex-dependence of the different adaptive mechanisms (a difference in thermal tolerance but not in phenotypic plasticity between the sexes) suggests that their physiological basis is different.

Multiple factors affect acute thermal stress tolerance. Understanding these factors, and how they vary among populations, has critical implications for predictions of future population dynamics. Lower male thermal tolerance creates an asymmetrical vulnerability to climate change, which could lead to

population declines under less intense warming due to sperm-limitation [27,55]. Our results also suggest that despite having a higher thermal tolerance, low-latitude populations may be more vulnerable to projected warming. That is, the small difference between $LD_{50}$ values and average annual maximum temperature indicates that copepods in these regions are near their thermal limit under present conditions. As they are also unable to respond to increased temperatures through developmental phenotypic plasticity, any further increase in temperature is likely to have deleterious effects on population survival, as previously suggested for tropical copepods [12,56]. Furthermore, males have a significantly lower thermal tolerance, further lowering the temperature threshold that would bring the onset of temperature-driven population dynamic changes. Both sexes in the CT population, however, have thermal tolerance values well above the current temperature maximum, and have a robust plastic capacity to increase thermal tolerance, potentially decreasing deleterious effects of warming on this population. Contemporary thermal tolerance and phenotypic plasticity are just two of the determinants of vulnerability to climate change. Rapid adaptation to changing climate may also affect population vulnerability [57,58]. In that regard, our results suggest that temperature-driven selective pressure may be different for the two sexes. The implications of this for the evolutionary dynamics of thermal adaptation are largely unexplored [59].

Ultimately, thermal tolerance and the mechanisms affecting it are only components of the suite of factors that will determine vulnerability to climate change. While genetic differentiation and phenotypic plasticity are two of the major adaptive mechanisms, behaviour may also play a large role in response to warming [60,61]. Avoidance of adverse conditions and range shifts are also possible, especially in planktonic organisms [62,63]. Further, there are ecological effects of warming such as changing patterns of primary productivity [64–66] as well as changes in predator–prey interactions [67–70]. Phenological mismatches between copepods, their prey and their predators, for example, could also have profound effects on marine communities [71,72]. These ecological factors could interact with warming directly; food availability has been shown to affect thermal tolerance in other organisms [73–76]. It is clear that multiple adaptive mechanisms and factors will determine vulnerability to climate change. We show here that among these factors, genetic differentiation, phenotypic plasticity and individual sex have significant, population-specific influences on thermal tolerance. Hence, they should be taken into consideration in a wider range of model systems.

Data accessibility. Sasaki MC, Hedberg S, Richardson K, Dam HG. Data from: Complex interactions between local adaptation, plasticity, and sex determine vulnerability to warming in a widespread marine copepod. Dryad Digital Repository: http://dx.doi.org/10.5061/dryad.v5g6r80.

Authors' contributions. H.G.D. and M.S. designed the study. M.S. collected and cultured all copepods. K.R. and S.H. performed the experiments. M.S. analysed the data. M.S. and H.G.D. drafted the manuscript. All authors read and approved the manuscript.

Competing interests. We have no competing interests.

Funding. Supported by NSF grants 1559180 and 1658663, a Research Council grant from the University of Connecticut, and graduate research fellowships from the Department of Marine Sciences, University of Connecticut, USA.

Acknowledgements. S.H. and K.R. were participants in the UCONN-Mystic Aquarium Research Experience for Undergraduates program (http://www.mysticaquarium.org/reu/). We thank Drs Tracy Romano and Michael Finiguerra for organizing and leading this programme.

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
