## [Reviewer comments · Royal Society Open Science]

Revised submission:	21 February 2019	
Final acceptance:	27 February 2019	

Review History

Decision letter (RSOS-182115.R0)

13-Feb-2019

Dear Mr Sasaki

On behalf of the Editors, I am pleased to inform you that your Manuscript RSOS-182115 entitled "Complex interactions between local adaptation, phenotypic plasticity, and sex affect vulnerability to warming in a widespread marine copepod" has been accepted for publication in Royal Society Open Science subject to minor revision in accordance with the referee suggestions. Please find the referees' comments at the end of this email.

The reviewers and handling editors have recommended publication, but also suggest some minor revisions to your manuscript. Therefore, I invite you to respond to the comments and revise your manuscript.

- Ethics statement

- Data accessibility

<http://datadryad.org/submit?journalID=RSOS&manu=RSOS-182115>

- Competing interests

- Authors' contributions

- Acknowledgements

- Funding statement

Please ensure you have prepared your revision in accordance with the guidance at <https://royalsociety.org/journals/authors/author-guidelines/> -- please note that we cannot publish your manuscript without the end statements. We have included a screenshot example of

the end statements for reference. If you feel that a given heading is not relevant to your paper, please nevertheless include the heading and explicitly state that it is not relevant to your work.

Because the schedule for publication is very tight, it is a condition of publication that you submit the revised version of your manuscript before 22-Feb-2019. Please note that the revision deadline will expire at 00.00am on this date. If you do not think you will be able to meet this date please let me know immediately.

Please note that Royal Society Open Science charge article processing charges for all new submissions that are accepted for publication. Charges will also apply to papers transferred to Royal Society Open Science from other Royal Society Publishing journals, as well as papers

submitted as part of our collaboration with the Royal Society of Chemistry (<http://rsos.royalsocietypublishing.org/chemistry>).

on behalf of Dr Punidan Jeyasingh (Associate Editor) and Kevin Padian (Subject Editor)
openscience@royalsociety.org

Associate Editor Comments to Author (Dr Punidan Jeyasingh):

Associate Editor

Comments to the Author:

This manuscript quantified thermal tolerance in a copepod, and the effects of genetics, sex, and developmental plasticity on thermal tolerance. This represents a substantial amount of work (even if it compares only two populations). I urge the authors to place this work in a broader context, and give a flavor for the complexity involved in this problem (of which they illuminated two - genetic variation, and plasticity). From the rebuttal to BioLett, I gather this indeed was the intent of the authors, so I was left wondering why the problem was given such a biased treatment.

Specific comments:

L71. Some more info about the species would be useful. Basic life history and cycle. Size of the sexes etc.

L82. Refer to the table showing these values.

L82. Define shallow. The following qualifying sentence implies that shallow enough that thermal stratification doesn't occur?

L87. unnecessary to open the drift can of worms...no data is presented to test it, better leave it out. Just give ranges (instead of >1500).

L88. Need more info on common garden. What media was used? How were algae grown? How much algae was fed? How often?

L88. what is psu?

L97. How was mortality verified?

L99. How many from each site? $x_{FL} + y_{CT} = 1717$. Rationalize the imbalance in sample size.

L100. I am surprised size wasn't included here. Given temperature-size rule and Bergman's rule potentially impacting size between sites, and associated size-dependent effects on fitness-relevant traits such as fecundity. If size differences were negligible between sites, that should be supported by data (I'd think FL will be smaller than CT - which would make the result of higher thermal tolerance even more impressive!). Regardless, this size issue needs to be

addressed/acknowledged somewhere in the ms. It doesn't take away from the study, only makes it more complete in my opinion.

L160. This may be true as the ultimate consequence, but there are several proximate alternatives the authors could discuss.

L190. Agree with the conclusion/recommendation. However, this manuscript, as written, seems to do the opposite (i.e. focus on only a few mechanisms for which the authors collected data). That's fine, but to be fair to readers, a broader discussion of other variables/mechanisms would improve the paper. Size (see above). Food quantity and quality (wouldn't FL have higher primary production than CT?), perhaps nullifying the observed effects in the lab (where food was constant) - i.e. higher energetic intake may help them mitigate thermal stress for longer. These sort of issues can be discussed in RSOS with more space compared to BioLett, and I encourage the authors to consider.

I look forward to a fresh version to make a final decision.

Author's Response to Decision Letter for (RSOS-182115.R0)

See Appendix A.

Decision letter (RSOS-182115.R1)

27-Feb-2019

Dear Mr Sasaki,

I am pleased to inform you that your manuscript entitled "Complex interactions between local adaptation, phenotypic plasticity, and sex affect vulnerability to warming in a widespread marine copepod" is now accepted for publication in Royal Society Open Science.

on behalf of Dr Punidan Jeyasingh (Associate Editor) and Professor Kevin Padian (Subject Editor)
openscience@royalsociety.org

Associate Editor Comments to Author (Dr Punidan Jeyasingh):

I thank the authors for addressing my comments. This version is much improved. I am happy to recommend it for publication.

Appendix A

Rebuttal Statement: Our original submission to Open Science received thoughtful and helpful comments from a reviewer, for which we are extremely grateful. We have made an effort to address the concerns, as outlined below. Overall, we agree with the reviewer that several topics needed to be included or more thoroughly discussed, and we have made an effort to expand the discussion as suggested. However, we have limited this expansion to keep the paper focused on its original goal: identifying factors that are commonly ignored in assessments of vulnerability to climate change.

Please find below a point-by-point response to concerns/suggestions from the original reviewer:

L71. Some more info about the species would be useful. Basic life history and cycle. Size of the sexes etc.

We have included this information in the introduction. Lines 70-76

L82. Refer to the table showing these values.

We have referred to the relevant table. Line 86

L82. Define shallow. The following qualifying sentence implies that shallow enough that thermal stratification doesn't occur?

We have provided this information. Line 88

L87. unnecessary to open the drift can of worms...no data is presented to test it, better leave it out. Just give ranges.

The statement about drift was deleted. New statement is in line 92.

L88. Need more info on common garden. What media was used? How were algae grown? How much algae was fed? How often?

We have provided this information. Lines 92-100.

L88. what is psu?

We have defined this term on lines 93-94.

L97. How was mortality verified?

We provide detailed information on determining mortality on line 115.

L99. How many from each site? xFL + yCT= 1717. Rationalize the imbalance in sample size.

We have provided this information and explained why different number of individuals were used from the two populations. Lines 115-120.

L100. I am surprised size wasn't included here. Given temperature-size rule and Bergman's rule potentially impacting size between sites, and associated size-dependent effects on fitness-relevant traits such as fecundity. If size differences were negligible between sites, that should be supported by data (I'd think FL will be smaller than CT - which would make the result of higher thermal tolerance even more impressive!). Regardless, this size issue needs to be addressed/acknowledged somewhere in the ms. It doesn't take away from the study, only makes it more complete in my opinion.

We have provided information on body size measurements (lines 102-106). We have also added in relevant results (lines 139-146) and discussion sections on this topic (lines 188-197 & lines 207-212).

L160. This may be true as the ultimate consequence, but there are several proximate alternatives the authors could discuss.

We added in discussion on how body size may affect thermal tolerance. However, we feel that in-depth discussion of other alternatives may be less relevant to the goal of the paper (to call attention to factors that are often ignored in assessments of vulnerability). As our data set is insufficient to strictly test these alternatives, we have refrained from undue speculation.

L190. Agree with the conclusion/recommendation. However, this manuscript, as written, seems to do the opposite (i.e. focus on only a few mechanisms for which the authors collected data). That's fine, but to be fair to readers, a broader discussion of other variables/mechanisms would improve the paper. Size (see above). Food quantity and quality (wouldn't FL have higher primary production than CT?), perhaps nullifying the observed effects in the lab (where food was constant) - i.e. higher energetic intake may help them mitigate thermal stress for longer.

We have addressed some of the other variables/mechanisms that may be important (lines XX-XX). The reviewer was right in asking for a more well-balanced coverage, but the assumption that patterns in primary productivity between sites could explain patterns does not apply for two reasons. I) Our common garden experimental design should remove any direct effects of differing amounts of primary productivity between sites. II) Primary productivity is actually higher in CT than in FL, rather than vice versa. Any remnant environmental acclimation might therefore have the opposite effect from what the reviewer suggested (it would decrease thermal tolerance in the FL population, not the CT population).